# The Use of Palmitoylethanolamide in the Treatment of Long COVID: A Real-Life Retrospective Cohort Study

**DOI:** 10.3390/medsci10030037

**Published:** 2022-07-14

**Authors:** Loredana Raciti, Rosaria De Luca, Gianfranco Raciti, Francesca Antonia Arcadi, Rocco Salvatore Calabrò

**Affiliations:** 1GCA-Centro Spoke AO Cannizzaro, Catania, IRCCS Centro Neurolesi Bonino Pulejo, 98123 Messina, Italy; loredana.raciti79@gmail.com (L.R.); gianfranco.raciti@irccsme.it (G.R.); 2Behavioral and Robotic Neurorehabilitation Unit, IRCCS Centro Neurolesi Bonino Pulejo, 98123 Messina, Italy; rosaria.deluca@irccsme.it (R.D.L.); francesca.arcadi@irccsme.it (F.A.A.)

**Keywords:** long COVID, persistent post-COVID syndrome, PEA, post-COVID-19 functional status scale

## Abstract

COVID-19 can cause symptoms that last weeks or months after the infection has gone, with a significant impairment of quality of life. Palmitoylethanolamide (PEA) is a naturally occurring lipid mediator that has an entourage effect on the endocannabinoid system mitigating the cytokine storm. The aim of this retrospective study is to evaluate the potential efficacy of PEA in the treatment of long COVID. Patients attending the Neurological Out Clinic of the IRCCS Centro Neurolesi Bonino-Pulejo (Messina, Italy) from August 2020 to September 2021 were screened for potential inclusion in the study. We included only long COVID patients who were treated with PEA 600 mg two times daily for about 3 months. All patients performed the post-COVID-19 Functional Status (PCFS) scale. Thirty-three patients (10 males, 43.5%, mean age 47.8 ± 12.4) were enrolled in the study. Patients were divided into two groups based on hospitalization or home care observation. A substantial difference in the PCFS score between the two groups at baseline and after treatment with PEA were found. We found that smoking was a risk factor with an odds ratio of 8.13 CI 95% [0.233, 1.167]. Our findings encourage the use of PEA as a potentially effective therapy in patients with long COVID.

## 1. Introduction

Coronavirus disease (COVID-19) is an infectious disease caused by the SARS-CoV-2 virus. Most people infected with the virus will experience mild to moderate respiratory illness and recover without requiring special treatment. However, some will become seriously ill and require medical attention. This could be due to either a multiple organ failure or severe lung dysfunction [1,2] with hypoxic encephalopathy and death [3,4,5,6]. In fact, in severe cases there is the so called “cytokine storms” involving the release of pro-inflammatory cytokines (e.g., interleukins 1, 6, 8, 17, and 1β), and tissue necrosis factor α (TNF-α) that contribute to the rapid systemic organ failure observed in select critically ill COVID-19 patients [6]. Moreover, thrombotic complications are caused by the SARS-CoV-2 outer surface important phospholipidic envelope that favors assembly of autoantibodies against this phospholipidic envelope (e.g., lupus anticoagulant, anti-cardiolipin, anti-β2glycoprotein), with a consequent serious antiphospholipid syndrome [7].

For some people, COVID-19 can cause symptoms that last weeks or months after the infection has gone, with a significant impairment of quality of life (QOL). This condition is defined as “persistent post-COVID syndrome” (PPCS) or “long COVID” [8]. The main symptoms include extreme tiredness, shortness of breath, chest pain or tightness and “brain fog” (that is problems with memory and concentration), as well as difficulty sleeping, heart palpitations, dizziness and joint pain. Persistence of anosmia and dysgeusia [9] or neuropsychiatric complications are also possible [10,11,12,13]. Therefore, the increasing importance of long COVID symptoms and burden, led the WHO organization to plan several international meetings with experts reaching a consensus on a definition and reporting details and the subtypes and case definitions of this condition [13].

The etiology of long COVID is represented by the protracted critical chronic immunosuppression condition defined [14,15] as persistent inflammation, immunosuppression, and catabolism syndrome [8]. Indeed, it is believed that the hyperactivation of the immune system with abnormalities in the coagulation system [16,17,18,19,20,21,22,23] could persist in predisposed individuals, although with a lower intensity causing the chronic symptomatology of long COVID.

To disable the pathogenicity of SARS-CoV-2, several anti-inflammatory products have been proposed. Many immunosuppressive drugs [24,25,26], such as chloroquine, hydroxychloroquine, favipiravir, monoclonal antibodies, antisense RNA, corticosteroids, convalescent plasma and vaccines are being evaluated [27]. Other products, as well as diet and nutrition for modulating immune function [6,28,29] and to increase host defenses against viral infections [18,19] have been used. However, conflicting information has been reported [30,31].

Recently, an exercise protocol for COVD-19 patients has been proposed to deal with long COVID symptoms [32] and involved patients with resolution of symptoms at least 2 weeks before enrollment in the case of severe forms, or 1 week in the moderate forms [17]. The protocol consisted of a screening part with scales and tests, and the rehabilitation session lasting 12 weeks and using daily aerobic exercises plus two sessions per week of resistance training.

On the other hand, when patients have difficulty following a physiotherapy training at the rehabilitation centers, the innovative methods of the telemedicine might be used either for psychological support (“tele-mental-health” or “cyberhealth psychotherapy”) [33,34,35,36], improving motivation and compliance of the patients to the motor training assessment [37], or for physical training.

Moreover, more attention has been yielded to endocannabinoid-related complexes and endogenous bioactive lipid amides [38,39] to account for pleiotropies and homeostatic responses regulating immune response, improving pain and suppression of inflammation [10,40,41,42]. Recently, palmitoylethanolamide (PEA), obtained from peanuts or fenugreek seeds and soybean lecithin [43], has been increasingly used in the SARS-CoV2 infection due to the capacity of preventing mast cell (MC)-induced pulmonary inflammation and fibrosis during SARS-CoV2 infection [44], reducing the cytokine storm and cell-mediated immunity [45,46,47].

We have thus hypothesized that an anti-inflammatory phytotherapy could be an alternative therapy for the longer persistence of the viral infection of COVID-19.

Therefore, aim of this retrospective study is to evaluate the potential efficacy of PEA in the treatment of long COVID.

## 2. Materials and Methods

This is a single-center open-label pre–post single-arm retrospective cohort study. Patients attending the Neurological Outclinic of the Piemonte Hospital, IRCCS Centro Neurolesi “Bonino-Pulejo” (Messina, Italy) from August 2020 to September 2021 and presenting with symptoms of long COVID were screened for potential inclusion in the study. The patients enrolled were referred by general practitioners or other specialists or they reached the clinic on their own because of the persistence of symptomatology.

The diagnosis of long COVID was posed according to the WHO definition: “Long-COVID occurs in individuals with a history of probable or confirmed SARS-CoV-2 infection, usually 3 months from the onset of COVID-19 with symptoms that last for at least 2 months and cannot be explained by an alternative diagnosis” [48]. Physical, neurological, as well as cardiorespiratory examinations were performed, and when necessary patients were submitted to instrumental investigation. Demographic and clinical characteristics were also collected (Table 1).

We included only long COVID patients who were treated with PEA, following an internal common protocol, i.e., 600 mg two times daily for about 3 months. All patients were administered the post-COVID-19 Functional Status (PCFS) scale, to assess functional sequelae, before (T0) and at the end of the treatment (T1, i.e., after 3 months). The PCFS is an ordinal scale ranking patients in meaningful categories, and it can be used to track improvement over time. The scale may be either self-reported or performed as a standardized interview [49,50].

Patients with autoimmune disorders, peripheral nerve pathology, rheumatological and preexistent cardiorespiratory problems, as well as those with severe psychiatric disorders were excluded from the study.

The study population was divided into two groups: hospitalized and home caring patients. Hospitalization involved the acute phase of COVID, and none of the patients were hospitalized during the treatment with PEA. For every group all demographic characteristics were examined, with regard to gender and smoking habits (see Table 2).

The Hospital Research Ethical Committee of the IRCCS Neurolesi approved this study (IRCCSME.CE 27/2021), informed consent for data publication was obtained by all participants.

### Statistical Analysis

The changes of the outcome measures were compared between the groups using a *t*-test, with the level of significance (alpha) set to 0.05. Continuous variable data were expressed as means ± SD, whereas categorical variables were expressed as frequencies and percentages. The χ2 test with continuity correction was used to assess the statistical differences in proportions. All the analyses were conducted according to an intention-to-treat analysis, including all participants for which data were available.

To assess the association between the consumption of PEA and the slowing or reversing of cognitive decline, we used Fisher’s exact test because our sample size was small, and we reported odds ratios (OR) and the associated 95% confidence intervals (CI).

## 3. Results

Thirty-three patients (10 male, 43.5%, and 23 female, 69.7%, mean age 47.8 ± 12.4) were enrolled in the study. All of them were treated with PEA (600 mg/daily) without any side effects. Each patient’s demographic and clinical features are reported on Table 1.

Patients were divided into two groups based on hospitalization or home care management. Hospitalized patients showed a higher mean age compared to home care patients, even though a statistically significant difference was not obtained. On the other hand, a substantial difference in PCFS score between the two groups at baseline and after treatment with PEA were found. The main features and difference of the two groups are reported in Table 2.

Separating patients into smoker (Sm) and non-smoker (Nsm) groups, 83% of hospitalized patients were smokers compared with home caring and no smokers with Fisher’s exact test statistic value < 0.00001. The result was significant at *p* < 0.05 and all the Sm hospitalized patients (HSm) were, as expected, treated with cortisone and antibiotics (or combined therapy), as well as heparin and non-invasive ventilation (NIV). None of the home care smoker (NhNsm) patients were, on the contrary, treated (see Table 3).

Moreover, the HSm group showed a lower degree of improvement as well as a higher grade of limitations with statistically significant differences in PCFS score at t0 (3.8 ± 0.4 vs. 2.6 ± 0.5) and t1 evaluation (2.2 ± 1.3 vs. 0.4 ± 0.5) with respect to NhNSm (see Table 4).

On the other hand, about 30% of NhNsm were treated with corticosteroids or antibiotics compared to Hsm patients. All patients completed the treatment with PEA without any side effects. Comparing the two groups (Sm and NSm), we evaluated the relative risk as the probability of the hospitalization occurring for the Sm group divided by the probability of it occurring for the Nsm group. We found a relative risk of 4.17 with an odds ratio of 8.13 CI 95% [0.233, 1.167]; this represents the odds that hospitalization will occur in Sm patients compared with Nsm.

## 4. Discussion

As far as we know, this is the first study showing the potential efficacy of PEA in improving symptomatology of patients with long COVID. Indeed, most of our patients obtained a significant improvement in PCFS score (*p* = 0.0000) after treatment with PEA and without any side effects. Therefore, a causal beneficial effect of this 3-month therapy seems to be a reasonable assumption in our population. However, our results cannot be generalized to the total population of COVID-19 patients, as a control group treated with placebo is missing. Moreover, a spontaneous recovery could not be totally ruled out, although this is not so probable as it happens in nearly all of the patients after the nutraceutical intake.

The efficacy of PEA could be explained by its anti-inflammatory properties as well as its antiviral activity.

The PEA, as an endogenous fatty acid amide, has anti-inflammatory action because of several complex mechanisms. Indeed, the compound has the ability to bind the peroxisome proliferator-activated receptor-α (PPAR-α), also known as nuclear receptor subfamily 1—group C—member 1 (NR1C1), [51], and to downregulate the mastocytes hyperactivity by the feedback mechanism of “autacoid local inflammation antagonism” (ALIA), that gives PEA the name of ALIAmide [52].

Another mechanism of anti-inflammation is the resemblance with the vanilloid receptor 1 (VR1) and the cannabinoid-like G-protein-coupled receptors 55 (GPR55) and 119 (GPR119) [53], reducing endocannabinoid catabolism and increasing their concentrations [54,55,56].

As an antagonist of the nuclear factor-κB (NF-κB) signaling pathway, PEA regulates the activation of the PPARα receptors [57,58], increasing macrophage activation and phagocytosis, without an increase in proinflammatory cytokines [57], regulating fatty acid metabolism [59,60] and reducing lipoperoxidation and reduction of nitric oxide by the downregulation of inducible nitric oxide synthase (iNOS) [61], and other signal mediators (such as aCOX2, S100B and GFAP) [62], as well as the modulation of the aberrant crosstalk between PPARα and TLRs [57,63].

All these mechanisms prevent the endothelial damage that accounts for the pathogenesis of the systemic inflammatory response in severe COVID-19 [64] and allows for the use of PEA in influenzas and other common colds [65], as well as for pain [66], neuroprotection and as an anti-convulsant molecule [67].

PEA has shown the capacity to decrease the expression and release of pro-inflammatory particles in experimental models and in cultured sections of dextran-sodium-sulfate-induced ulcerative colitis (UC) [63,64,65,66,67,68,69,70].

The neuroprotective and anti-inflammatory effect of PEA has been recently studied in patients with smell dysfunction. In a recent clinical trial on the efficacy of PEA on olfactory dysfunction, it has shown that a combination of PEA and luteolin (PEA-LUT) with olfactory training was more efficient in recovering smell than olfactory training alone [71].

Therefore, based on the theory of chronic inflammation in long COVID and immunological involvement in chronic fatigue syndrome, we showed that PEA could be used to resolve inflammatory processes, reducing the progression of chronic inflammation and solving the chronic pain and fatigue in PPCS, as previously demonstrated [10,72].

Moreover, in the case of asymptomatic persistence of the virus, PEA has been shown to have antiviral mechanisms [73]. Indeed, PEA disassembles lipid droplets, avoiding the fonts of energy and defense by SARS-CoV-2 against innate cellular defenses, thanks to the activation of PPAR-α. Then, as the persistence of low-level virus particles/mRNA can be detected for long time after clinical recovery, this further mechanism of action can help in treating long COVID.

The lack of drug interactions and absence of adverse effects of PEA is an important advantage in the management and adherence to the therapy, also in the earliest stages of COVID-19 [74].

An important finding in our study is the evidence of the differences between hospitalized and home care patients around COVID symptoms, as hospitalized patients showed statistically significant differences in PCFS score compared with home care patients. Moreover, we obtained evident differences between smoker and non-smoker hospitalized and home care patients, with the HSm groups showing a lower degree of improvement as well as a higher level of limitations with statistically significant differences in PCFS scores.

Notably, our data are in line with a meta-analysis from 2020 that showed how smokers had higher odds (1.91 times) of COVID-19 progression than never smokers [75].

In our study we found an even higher OR (8.13) for hospitalization that will occur in Sm patients compared with NSm patients. Moreover, our data confirmed that patients with COVID-19 still have high levels of fatigue and anhedonia after recovery from infection [76], especially in COVID-19 patients discharged from hospital. These patients usually assert that they were unable to return to their baseline activity level due to fatigue [76].

The study has some limitations. First, the retrospective design that prevented us from making any a priori hypothesis. The relatively small sample size and the absence of a control group (taking other drugs or placebo) are other limitations. Moreover, we evaluated the concentration of VES and C-reactive protein only in some patients; indeed, the concentration in the entire sample could help to more objectively determine the potential effect of PEA on inflammation. However, this is an open-label observational study, without an a priori hypothesis, performed in a real-life context that could be the basis for planning future randomized clinical trials.

## 5. Conclusions

Our findings, although coming from a retrospective open-label study, suggest the idea that PEA might be an effective therapy in patients with long COVID. However, further preclinical and clinical trials are necessary to evaluate long-term efficacy of PEA as a promising adjuvant therapy in this syndrome that negatively affects quality of life.

## Figures and Tables

**Table 1 medsci-10-00037-t001:** Long COVID Patients’ demographic and clinical characteristics.

Pts	Age	G	H	LH (Days)	SL (Months)	CMD	Drugs	PCFS t0	PCFS t1	Smokers	CPAP	Cortisone	Antibiotics	Heparin
1	45	f	no		7			3	0	No	No	No	No	No
2	58	m	yes	10	8	DM	Metformine	4	2	Yes	Yes	Yes	Yes	Yes
3	39	f	no		5			3	0	No	No	Yes	No	No
4	65	f	yes	15	7	Hyp	Ramipril	4	2	Yes	Yes	Yes	Yes	Yes
5	28	m	no		5	CM	Topiramate	3	0	No	No	Yes	Yes	No
6	55	f	no		3	Dep	Venlafaxine	3	1	Yes	No	No	No	No
7	44	f	no		4	CM		3	0	Yes	No	No	No	No
8	54	m	no		7			3	0	No	No	No	No	No
9	60	m	yes	21	6	OB, DM, Hyp	Captopril, Metformin	4	4	Yes	Yes	Yes	Yes	Yes
10	55	f	no		8	Hyp	Analapril	3	1	No	No	No	Yes	No
11	22	f	yes	7	5			4	0	Yes	No	Yes	Yes	Yes
12	29	f	no		6			3	0	No	No	Yes	No	No
13	33	m	no		8	DM	Insulin	3	1	No	No	No	No	No
14	49	m	no		7			3	0	Yes	No	No	No	No
15	63	m	no		7			2	0	Yes	No	No	No	No
16	39	f	no		5			3	1	No	No	Yes	No	No
17	55	f	yes	12	6	CM		4	2	No	No	Yes	Yes	Yes
18	48	f	no		9			3	0	Yes	No	No	No	No
19	64	f	yes	10	9	Hyp	Amlodipine	2	2	No	No	Yes	Yes	Yes
20	59	m	no		4			2	0	Yes	No	No	No	No
21	61	f	yes	8	7			4	1	Yes	No	Yes	Yes	No
22	58	f	yes	18	12	OB, RA		4	4	Yes	Yes	Yes	Yes	Yes
23	47	f	no		7	CM	Flunarizine	3	1	Yes	No	No	No	No
24	56	f	no		11			3	0	No	No	No	Yes	No
25	42	m	yes	21	10			4	1	Yes	Yes	Yes	Yes	Yes
26	54	f	no		10			3	1	No	No	No	No	No
27	47	f	no		7			3	0	No	No	No	No	No
28	21	f	no		7			2	0	No	No	No	Yes	No
29	27	f	yes	15	8			3	2	Yes	Yes	Yes	Yes	Yes
30	58	f	yes	15	6	DM, Hyp	Metformin; Captopril	3	3	Yes	Yes	Yes	Yes	Yes
31	41	f	no		9			2	1	Yes	No	No	No	No
32	52	f	yes	18	5			4	3	Yes	No	Yes	Yes	Yes
33	51	m	no		8			3	1	No	No	No	No	No

Pts: patients; G: gender; H: hospitalization; LH: length of hospitalization; SL: length of symptoms before PEA treatment; CMD: comorbidity; DM: diabetes mellitus; Hyp: hypertension; CM: chronic migraine; OB: obesity; Dep: depression; RA: rheumatoid arthritis; PCFS: post-COVID-19 functional status at the time of observation (PCFS t0) and the PCFS score at the follow-up (PCFS t1); C-PAP: continuous positive airway pressure.

**Table 2 medsci-10-00037-t002:** Clinical features of outpatients’/home care and hospitalized patients treated with PEA.

	Total (*N* = 33)	Home Care (*N* = 21)	Hospitalized (*N* = 12)	*p*
Age	47.8 ± 12.4	42.6 ± 14	51.8 ± 13.5	0.08
Women	23	14 (66.7%)	9 (75%)	0.25
Symptoms length (months)	7.1 ± 2	7.1 ± 2	7.4 ± 2	0.68
Comorbidity	13	6	7	
Diabetes	6	1	5	
Hypertension	5	1	4	
Obesity	2	0	2	
Migraine	4	3	1	
Depression	1	1	0	
Smoke	18	8 (38.1%)	10 (83.3%)	
PCFS * T0	3.12 ± 0.65	3 ± 0.65	3.7 ± 0.6	0.000
PCFS * T1	1.03 ± 1.19	1.03 ± 1.19	2.2 ± 1.1	0.01
Statistical difference PCFS T0–PCFS T1	*p* = 0.0000	*p* = 0.0000	*p* = 0.0004	
Symptoms improvement (SI) ^	2.03 ± 1.05	2.03 ± 1.05	1.1 ± 1.0	0.02

* PCFS: post-COVID-19 functional status at the time of observation (PCFS T0) and the PCFS score at follow-up, i.e., after 3 months (PCFS T1). ^ The symptom improvement (SI) expressed if symptoms disappeared (3+), remarkably improved (2+), improved (+) or unchanged (=). A categorical number from 0 (unchanged) to 3 has been arranged.

**Table 3 medsci-10-00037-t003:** PEA treatment differences between (Sm) and non-smokers (Nsm), hospitalized vs. home care.

Tot (*N* = 33)	Home Care (*N* = 21)	Hospitalized (*N* = 12)	*p*
Smokers	8 (38.1%)	10 (83.3%)	0.01
Cortisone (CO)	0	10	0.003
Antibiotics (An)	0	10	0.003
An + CO	0	10	0.06
Heparin	0	9	NV
CPAP	0	7	NV
Non-smokers	13 (61.9%)	2 (16.7%)	0.01
Cortisone	4	2	0.003
Antibiotics	4	2	0.003
Antibiotics + cortisone	1	2	0.06
Heparin	0	2	NV
C-PAP *	0	0	NV

* continuous positive airway pressure (C-PAP). The results were obtained by *t*-test analyses.

**Table 4 medsci-10-00037-t004:** Clinical Characteristics of HSm and NhSm groups treated with PEA.

Smokers	Home Care (*N* = 8)	Hospitalized (*N* = 10)	*p*
Age	50.8 ± 7.6	50.3 ± 15	0.9
SI ^	2.50 ± 0.76	1.10 ± 0.99	0.005
Symptom length	6.3 ± 2.3	7.4 ± 2.2	0.3
PCFS * t0	2.6 ± 0.5	3.8 ± 0.4	0.0000
PCFS * t1	0.4 ± 0.5	2.2 ± 1.3	0.002
Gender (M)	3 (37.5%)	3 (30%)	0.73

* PCFS: post-COVID-19 functional status. NhSm: non-hospitalized smoker patients. ^ The symptom improvement (SI).

## Data Availability

Data are available on demand to the corresponding author.

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
