# Peer review of "The Use of Palmitoylethanolamide in the Treatment of Long COVID: A Real-Life Retrospective Cohort Study"

_medsci, 2022, doi:10.3390/medsci10030037_

Round 1

Reviewer 1 Report

To the Authors:

General comments:

The authors evaluated the potential efficacy of palmitoylethanolamide (PEA) in the treatment of Long COVID in the study.  They found that the differences between hospitalized and home-care patients about COVID symptoms, as hospitalized patients showed a statistically significant differences in PCFS score as compared to home-care patients.  They concluded that their findings encourage the use of PEA as a potentially effective therapy in patients with Long COVID.  However, it is difficult to conclude whether PEA administration was directly associated with the improvement of Long COVID according to the findings from this study.

Specific comments:

1. As authors pointed out in the limitation section, this study does not have the control group. Based on meta-analysis or other reports, there are cases of Long COVID that can be cured over time without any specific treatment.  Therefore, the authors cannot conclude that PEA is effective for its treatment or not from the present study.  Only what we can discuss in this study is the difference between hospitalized patients and home-cared patient.

2. Why is it necessary to stratify regarding the presence or absence of hospitalization to discuss the efficacy of PEA?  If the authors intend to discuss the effectiveness of off-label use of a new drug, they should mention the details of individual cases, including side effects, dosing regimens, and other contemporaneous drugs used, for example, in the form of a case series.

3. The authors speculated that the efficacy of PEA could be explained by its anti-inflammatory properties.  Cortisone used for the patients with Long COVID is considered as a confounding factor.  Please mention this issue.

4. How were the laboratory markers for inflammation such as C-reactive protein or erythrocyte sedimentation rate in the clinical course?

Minor:

1. Abbreviations should not be used in the Title.

2. Since this study is in accordance with the WHO definition (post COVID-19 condition), it should be mentioned also in the introduction section.

3. There are two Tables 1 in pages 3 and 4.

4. In Table 1, "tot" is maybe a typo.

5. What did the P-values at the bottom of the table indicate?

6. Regarding the presence or absence of Hospitalization, is it during the acute phase of COVID treatment or during the period of PEA administration?  Please clarify it. 

7. In Table 2, "61.9%%" is also a typo.

8. In the text, the P-value comparing smokers to nonsmokers is described as < 0.00001, but in Table 2 it is shown as 0.01.  What does this discrepancy mean?

9. The authors obtained PCFS twice: t0 and t1.  Please specify what t1 means.

10. "NhSm" of the title of Table 3 seems also be a typo.  "Pcfs" in Table 3 is conceivably supposed to be "PCFS"?

Author Response

To rew 1.

General comments:

The authors evaluated the potential efficacy of palmitoylethanolamide (PEA) in the treatment of Long COVID in the study.  They found that the differences between hospitalized and home-care patients about COVID symptoms, as hospitalized patients showed a statistically significant differences in PCFS score as compared to home-care patients.  They concluded that their findings encourage the use of PEA as a potentially effective therapy in patients with Long COVID.  However, it is difficult to conclude whether PEA administration was directly associated with the improvement of Long COVID according to the findings from this study.

We agree with this comment, and tuned down conclusion.

Specific comments:

  1. As authors pointed out in the limitation section, this study does not have the control group. Based on meta-analysis or other reports, there are cases of Long COVID that can be cured over time without any specific treatment.  Therefore, the authors cannot conclude that PEA is effective for its treatment or not from the present study.  Only what we can discuss in this study is the difference between hospitalized patients and home-cared patient.

Thank you for your interesting observations. We agree with the reviewer that several cases of Long COVID were not treated. However, this condition, as defined by the WHO, is characterized by a mild to moderate symptoms that, even though usually not treated, impact on everyday functioning and still need to be better understand. This may be the reason why a specific therapy is still under debate and missing, and why we pointed out to better evaluate the efficacy of a known and worldwide anti-inflammatory therapy to identify strategies to improve long-term outcomes. We have better specified that most of the patients in this cohort improved, especially those hospitalized that had worse symptoms. We conclude that it might be effective, but RCT with a placebo control group are needed to confirm.

We added the following sentences: “As far as we know, this is the first study showing the potential efficacy of PEA in improving symptomatology of patients with Long-Covid. Indeed, most of our patients, either hospitalized or not, obtained a significant improvement in the PCFS score (P= 0.0000) after treatment with PEA. Therefore, a causal beneficial effect of this 3 month- therapy seem to be a reasonable assumption in our population. However, our results cannot be generalized to the total population of COVID-19 patients, as a control group treated with placebo is missing. Moreover, a spontaneous recovery could not be totally ruled out, although this is not so probable as it happens in nearly all the patients after the nutraceutical intake.”

  1. Why is it necessary to stratify regarding the presence or absence of hospitalization to discuss the efficacy of PEA?  

Thank you for your interesting observation. The ‘Post-COVID-19 Functional Status (PCFS) Scale’ was proposed as an easy tool to recognize people with incomplete recovery from COVID-19 infection, giving information on quality of life and functioning after infection. Severe symptoms leading to hospitalization due to COVID-19, of which some develop long-term sequelae. On the other hand, several individuals with mild COVID-19 experience bothering prolonged symptoms. Therefore, we assumed that stratify in hospitalized and home care was fundamental to evaluate the differences between severe and mild/low-risk Covid-19 patients and the difference of assessing health-related quality of life, daily life, and work impairment.

  1. If the authors intend to discuss the effectiveness of off-label use of a new drug, they should mention the details of individual cases, including side effects, dosing regimens, and other contemporaneous drugs used, for example, in the form of a case series.

Thank you for your comments and interesting observations. All patients were treated with 2 sachets die for about 3 months and no one have reported side effects, as better specified.

  1. The authors speculated that the efficacy of PEA could be explained by its anti-inflammatory properties.  Cortisone used for the patients with Long COVID is considered as a confounding factor.  Please mention this issue.

Thank you for your comment. However, the use of cortisone was made only during the acute Covid phase, mainly in the hospitalized patients. During the use of PEA patients did not take cortisone.  

  1. How were the laboratory markers for inflammation such as C-reactive protein or erythrocyte sedimentation rate in the clinical course?

Thank you for your comment. We evaluated the dosage of VES and C-reactive protein only in some patients, as this was a retrospective study and no a priopri methodology and hypothesis was done. Then, the data, although potentially interesting were missing. We added in limitation.

Minor:

  1. Abbreviations should not be used in the Title.

Thank you for your recommendation. We modified the Title as suggested

  1. Since this study is in accordance with the WHO definition (post COVID-19 condition), it should be mentioned also in the introduction section.

Thank you for your interesting observations. We added the follow sentences in the introduction section.

“Therefore, the increasing importance of long-Covid symptoms and burden, lead the WHO organization to plan several international meetings with experts reaching a consensus on a definition and reporting details and the subtypes and case definitions a of this condition [52]”.

  1. There are two Tables 1 in pages 3 and 4.

We added a new table 1 and re-ordered the others

  1. 4. In Table 1, "tot" is maybe a typo.

Yes, corrected

  1. What did the P-values at the bottom of the table indicate?

Thank you for your observations (nr 3, 4 and 5 comments). We modified and clarified the signed parts. The P-values at the bottom of the table 1 indicate the statistical differences between the PCFS score at the time of observation and the PCFS score at the follow-up (PCFS t1)

  1. Regarding the presence or absence of Hospitalization, is it during the acute phase of COVID treatment or during the period of PEA administration?  Please clarify it. 

We thank you for the comment. It was during the acute phase of COVID.

  1. In Table 2, "61.9%%" is also a typo.

Corrected

  1. In the text, the P-value comparing smokers to nonsmokers is described as < 0.00001, but in Table 2 it is shown as 0.01.  What does this discrepancy mean?

We thank for the comment and the good suggestions (comments nr 7,8). On table 2 was reported the differences of numerical values (8 home care smokers vs 10 hospitalized smokers’ patients), meanwhile in the text we reported the percentage of hospitalized (83.3%) and home care (38.1%) smokers, calculating the difference by the Fisher test.

  1. The authors obtained PCFS twice: t0 and t1.  Please specify what t1 means.

Thank you very much for your observation. T0 and T1 represents the two time point of the visit observation before starting PEA (t0) and at 3 months of treatment (t1). We added the “t1” in the methods section.

  1. "NhSm" of the title of Table 3 seems also be a typo.  "Pcfs" in Table 3 is conceivably supposed to be "PCFS"?

Thank you for your suggestions. We modified the text as suggested and added the definition of “NhSm”, that we meant as no hospitalized smokers’ patients.

Reviewer 2 Report

In this manuscript, the authors intend to study the use of palmitoylethanolamide (PEA) in the treatment of Long-Covid. They claimed that their findings encourage the use of PEA as a potentially effective therapy in patients with Long-Covid.

 Several suggestions:

1.      Two identical Table 1s are present in this manuscript. No use PEA in the treatment of long-Covid was presented in any of the three tables in the manuscript. Therefore, I don’t know whether the use of PEA was an effective therapy in patients with Long-Covid.

2.      Please check the entire manuscript. There are so many typo errors. For example, line 17, (10 males, 43.5%, mean age 47.8 ± 12,4). [43.5%] is not correct; [12.4] not [12,4]. Also in line 129. Line 81, [Based on all this information]. [Antibiotici] and [%%] in Table 2.

3.      Line 140-141, [compared to home caring and no smokers with the Fisher exact test statistic value < 0.00001], please check. Are there 8 smokers in the group of home caring?

4.      Table 3, how to get p [0.0000] in Pcsf*T0?

Author Response

TO Reviewer 2.

In this manuscript, the authors intend to study the use of palmitoylethanolamide (PEA) in the treatment of Long-Covid. They claimed that their findings encourage the use of PEA as a potentially effective therapy in patients with Long-Covid.

 Several suggestions:

  1. Two identical Table 1s are present in this manuscript. No use PEA in the treatment of long-Covid was presented in any of the three tables in the manuscript. Therefore, I don’t know whether the use of PEA was an effective therapy in patients with Long-Covid.

We thank for the comment and the good suggestions. We deleted the first table 1 and specify PEA therapy in the title of all the tables.   

  1. Please check the entire manuscript. There are so many typo errors. For example, line 17, (10 males, 43.5%, mean age 47.8 ± 12,4). [43.5%] is not correct; [12.4] not [12,4]. Also, in line 129. Line 81, [Based on all this information]. [Antibiotici] and [%%] in Table 2.

Thank you for your observations. We modified the text as suggested.

  1. Line 140-141, [compared to home caring and no smokers with the Fisher exact test statistic value < 0.00001], please check. Are there 8 smokers in the group of home caring?

We thank for the comment and the good suggestions. We performed a double analysis: on table 2 the difference was evaluated for the numerical values (8 home care smokers vs 10 hospitalized smokers’ patients), meanwhile in the text we reported the percentage of hospitalized (83.3%) and home care (38.1%) smokers, calculating the difference by the Fisher test.

  1. Table 3, how to get p [0.0000] in Pcsf*T0?

Thank you for your observation. We obtained that the PCFS score was statistically significant difference between the home care and the hospitalized smokers’ patients at the time of the first observation (t0).

Round 2

Reviewer 1 Report

The authors answered to all our comments.  The lack of appropriate control is a problem to conclude the clinical data.  We expect a future RCT study to conclude the effectiveness of PEA on the long COVID patients.

Reviewer 2 Report

In this revised manuscript, the authors have addressed the questions I have raised previously.